# Time-Course Transcriptome Analysis of *Aquilegia vulgaris* Root Reveals the Cell Wall’s Roles in Salinity Tolerance

**DOI:** 10.3390/ijms242216450

**Published:** 2023-11-17

**Authors:** Yun Bai, Haihang Yu, Lifei Chen, Yuan Meng, Yanmei Ma, Di Wang, Ying Qian, Dongyang Zhang, Xiaoyu Feng, Yunwei Zhou

**Affiliations:** College of Horticulture, Jilin Agricultural University, Changchun 130118, China; yunb@jlau.edu.cn (Y.B.); yuhaihang1117@163.com (H.Y.); lfchen@jlau.edu.cn (L.C.); yuanm1120684462@163.com (Y.M.); 18543465877@163.com (Y.M.); 17767743128@163.com (D.W.); qiany9071@163.com (Y.Q.); hxxyzdy@126.com (D.Z.); 18991342829@163.com (X.F.)

**Keywords:** *Aquilegia vulgaris*, salt stress, cell wall, transcriptome, transcription factor

## Abstract

Salt stress has a considerable impact on the development and growth of plants. The soil is currently affected by salinisation, a problem that is becoming worse every year. This means that a significant amount of salt-tolerant plant material needs to be added. *Aquilegia vulgaris* has aesthetically pleasing leaves, unique flowers, and a remarkable tolerance to salt. In this study, RNA-seq technology was used to sequence and analyse the transcriptome of the root of *Aquilegia vulgaris* seedlings subjected to 200 mM NaCl treatment for 12, 24, and 48 h. In total, 12 *Aquilegia vulgaris* seedling root transcriptome libraries were constructed. At the three time points of salt treatment compared with the control, 3888, 1907, and 1479 differentially expressed genes (DEGs) were identified, respectively. Various families of transcription factors (TFs), mainly AP2, MYB, and bHLH, were identified and might be linked to salt tolerance. Gene Ontology (GO) analysis of DEGs revealed that the structure and composition of the cell wall and cytoskeleton may be crucial in the response to salt stress. Kyoto Encyclopedia of Genes and Genomes (KEGG) analysis of the DEGs showed a significant enrichment of the pentose and glucuronate interconversion pathway, which is associated with cell wall metabolism after 24 and 48 h of salt treatment. Based on GO and KEGG analyses of DEGs, the pentose and glucuronate interconversion pathway was selected for further investigation. AP2, MYB, and bHLH were found to be correlated with the functional genes in this pathway based on a correlation network. This study provides the groundwork for understanding the key pathways and gene networks in response to salt stress, thereby providing a theoretical basis for improving salt tolerance in *Aquilegia vulgaris*.

## 1. Introduction

Soil salt stress is among the most significant abiotic challenges that plants encounter [1,2]. Salt stress is a serious problem in plant cultivation and production, with plants exposed to high salt concentrations suffering damage such as osmotic stress, oxidative damage, and ion toxicity, which ultimately affects their productivity, ornamental value, and practical applications [3]. Over time, plants have evolved several ways to cope with high-salinity environments, including physiological mechanisms that respond to salt stress and directly mitigate salt damage, as well as molecular mechanisms that regulate such physiological processes [4]. Quantitative trait locus (QTL) mapping, genome-wide association studies, and RNA sequencing analysis have been employed to investigate the molecular mechanisms underlying plant salt tolerance [5]. Of these methods, RNA sequencing analysis is proving to be a powerful tool for elucidating the mechanism by which plants tolerate abiotic stress [6,7]. For example, RNA-seq transcriptome data aided in uncovering the salt stress response molecular network in the saline plant *Lycium barbarum* and identifying crucial genes involved in the salt stress response [8]. Moreover, the use of time-course transcriptome analysis facilitates the acquisition of gene expression profiles at various stress treatment intervals. This approach provides critical insight into the process of a plant’s dynamic response to external environmental changes and is an efficient method for studying the gene regulatory network [9]. By analysing the time-course transcriptomes of *poplar* under salt stress, one study confirmed the regulatory mechanism in *poplar* during salt stress and revealed potential regulatory relationships between transcription factors (TFs) and salt stress response mechanisms [10].

When subjected to salt stress, plants develop intricate regulatory networks and adaptive mechanisms, such as molecular regulation, signal transduction, and substance metabolism [11]. Plants detect signals produced by external stimuli through signal transduction pathways via receptors, and then initiate a cascade of genetic responses to abiotic stresses. In recent years, several studies have shown that plant cell walls also play a role in the perception and response to salt stress. Plant cell walls not only provide protection against biotic and abiotic stresses, but are also the first organelle to perceive and respond to environmental challenges [12,13]. Under non-biological stress such as salt stress, the plant cell wall mainly undergoes changes in its structure and composition. Usually, plant cells have dedicated systems to monitor the functional integrity of and changes in the cell wall’s constituents [14]. Some of these mechanisms also induce the repair of damaged cell walls through processes such as cell wall metabolism, cytoskeleton rearrangement, vesicle transport, etc. [15,16]. In the regulation of cell walls, certain TFs may partake in the synthesis and alteration of plant cell walls by means of regulating genes downstream or forming complexes [17].

TFs may regulate downstream gene expression by interacting with other proteins to activate or inhibit the transcription of target genes, which enhances their resilience to adverse conditions [18,19]. Various studies have demonstrated that AP2, MYB, and bHLH genes widely participate in regulating the salt response. In *Arabidopsis*, *AtMYB42* regulates the expression of *SOS2*, leading to an uplift of salt tolerance, where overexpressing lines exhibit greater salt tolerance than the wild type, while *atmyb42* mutants demonstrate salt sensitivity [20]. Furthermore, bHLH transcription factors regulate *AtNHX*, which modifies salinity responses [21]. *TaDREB3* is used to enhance *Arabidopsis*’ tolerance to high-salt conditions by regulating stress genes, including *LEA7*, *RD29A*, *RD17A*, *RAS1*, and *HSP70* [22].

*Aquilegia vulgaris* is a species of perennial herb in the genus *Aquilegia* of the Ranunculaceae family. Known for its striking leaves and unique flowers, it is easy to hybridise, has good salt tolerance, and can withstand cold temperatures as it can overwinter unprotected in north-eastern China. Chen et al. found that *Aquilegia* plants have salt tolerance through a salt stress test performed on the plants, and Chen et al.’s transcriptome analysis of *Aquilegia vulgaris* leaves under salt stress revealed the response mechanism to this stress [23,24]. Currently, research on *Aquilegia* plants primarily focuses on flower spacing and colour, phylogeny, and floral genetics [25,26,27]. However, few studies have examined their salt resistance, and the investigation of the transcriptomic molecular mechanisms of *Aquilegia* plants remains incomplete. At the same time, the majority of investigations on *Aquilegia* have focused solely on leaves, with a dearth of analyses on alternative tissues, particularly roots. The roots of plants are directly exposed to soil salts, making root growth and development critical for plant survival under salt stress [28]. In this study, RNA-Seq technology was used to investigate the transcriptome of the root system of *Aquilegia vulgaris* seedlings exposed to NaCl stress at different time intervals (12 h, 24 h, and 48 h) to elucidate the transcriptome response to salt stress and to identify genes associated with salt tolerance. This study aims to improve our understanding of the molecular mechanisms underlying salt tolerance in *Aquilegia vulgaris* and to develop novel salt-tolerant *Aquilegia vulgaris* varieties by using genetic engineering approaches to create new germplasm with improved salt tolerance. The research findings also hold tremendous importance for the promotion and utilisation of *Aquilegia vulgaris*.

## 2. Results

### 2.1. Results of Transcriptome Sequencing and Assessment of Quality

RNA-seq technology was employed to analyse the samples through transcriptome sequencing at various intervals of salt treatment (Appendix A). A total of 12 RNA-seq libraries were established, which contained 84.18 Gb of clean data. Each sample resulted in over 6.62 Gb of clean data, with over 44,157,640 clean reads detected from each sample via the detection and filtration of raw reads. Furthermore, the GC content of each sample ranged from 40.98 to 41.71. The percentage of Q20 bases was equal to or greater than 97.90%, and the percentage of Q30 bases was 93.64% or higher. These results indicate high accuracy in base identification. The R^2^ value from the three biological replicates surpasses 0.90 (Appendix A), and the percentage of samples from individual sources compared to the genome exceeds 87% (Appendix A), indicating close alignment of the transcriptome data with the reference species. The Unigene assembly comprises a total of 28,088, where the highest number of lengths falls within the range of 1001 to 2000 bp (Appendix A). In conclusion, the transcriptome sequencing data obtained from the given sample have been of high quality. These data can meet the requirements of subsequent analysis and serve as a guarantee and basis for screening and analysing the accuracy of related salt tolerance genes.

### 2.2. Investigation of the Differentially Expressed Genes (DEGs)

A total of 7274 DEGs were identified using a more balanced expression between upregulated and downregulated genes compared to the control (Figure 1). Among these, 1879, 1070, and 713 genes were upregulated after 12, 24, and 48 h of salt stress. At the same time points, 2009, 837, and 766 genes were downregulated, respectively. At the 12 h mark, the number of DEGs was at its highest, and it gradually decreased as the salt stress time increased. The DEGs at the three time points intersected with each other, but there were also individual DEGs at each time point. There were a total of 514 DEGs that changed in all three time points, which may play specific roles in the early and late stages of salt stress experienced by the *Aquilegia vulgaris* seedlings.

### 2.3. Analysis of Expression and Enrichment of TFs among DEGs

Numerous TFs exhibit specific expression during salt stress treatment. To identify TFs that may be regulated by salt stress in the transcriptome of *Aquilegia vulgaris* root, we compared sequences with the help of the *Arabidopsis* database on PlantTFDB. In the roots, 242 DEGs were discovered as TFs, which could be classified into 37 families of TFs (Figure 2). Most TFs belong to the AP2, MYB, bHLH, NAC, and bZIP families. The biggest group of TFs was AP2 (33), with MYB (32) and bHLH (21) following closely behind. The analysis of expression patterns in the AP2, MYB, and bHLH families showed that within the AP2 family, 24 DEGs were upregulated after 12, 24, and 48 h of salt treatment, whilst 9 DEGs were downregulated during the same time points, when compared to 0 h. In the MYB family, a total of 24 DEGs exhibited upregulation in expression during all salt treatment periods, while 8 DEGs showed downregulation in expression during all salt treatment periods. As for the bHLH family, 13 DEGs were upregulated and 8 DEGs were downregulated throughout all salt treatment periods. Most of the DEGs in the AP2, MYB, and bHLH families displayed an upward trend in expression.

### 2.4. Analysis of Differences in Alternative Splicing (DAS)

AS events were identified by using the rMATs 4.0.1 (University of California, Los Angeles, CA, USA) software to characterise the five splicing patterns, such as Skipped exon (SE), Mutually exclusive exon (MXE), Alternative 5′ splice site (A5SS), Alternative 3′ splice site (A3SS) and Retained intron (RI). A total of 307 DAS events were identified in the three datasets (such as 123 SE, 78 RI, 28 A5SS, 58 A3SS, and 20 MXE) under salt stress (Figure 3). The DAS types of the three treatment groups were mainly dominated by SE. The largest count of DAS events observed after 12 h of salt treatment was 145, with a minimum count of 61 DAS events at 24 h. Additionally, 101 DAS events occurred after 48 h of treatment. It is worth noting that the number of DAS events at 48 h increased compared to 24 h, suggesting that prolonged salt stress treatment may lead to an increased number of AS events.

### 2.5. Enrichment Analysis of Gene Ontology (GO) Terms for DEGs

To gain a deeper understanding of the potential mechanism of response to salt stress in *Aquilegia vulgaris*, the GO function was annotated using DEGs between salt treatment and control at various time points. Subsequently, the first 30 GO terms were analysed statistically based on the enrichment significance of different treatments. GO enrichment analysis of these annotated DEGs shows that the DEGs participated in three categories: “biological process (BP)”, “cell composition (CC)”, and “molecular function (MF)” (Figure 4). By merging time points, “DNA replication” was significantly enriched in the BP category. The CC category showed significant enrichment in “cell wall”, “external encapsulating structure”, “chromosome”, “chromosomal part”, “extracellular region”, and “apoplast”. The MF category was significantly enriched in “protein heterodimerisation activity”, “microtubule binding”, and “cytoskeletal protein binding”. The results suggest that the structure and composition of the cell wall and cytoskeleton play a crucial role in the response to salt stress.

### 2.6. Enrichment Analysis of Kyoto Encyclopedia of Genes and Genomes (KEGG) Pathways for DEGs

To determine the metabolic pathways associated with salt tolerance and investigate the DEGs implicated in various metabolic pathways, all the metabolic pathways that the DEGs were involved in under different treatments were analysed and annotated with the KEGG database. Following salt stress for 12, 24, and 48 h, 989, 407, and 324 DEGs were classified into 118, 108, and 99 KEGG pathways, respectively. In total, there were 119 KEGG pathways, excluding the crossed pathway. Taken together across all time points, the *Aquilegia vulgaris* root DEGs genes were mainly enriched (Figure 5) in DNA replication, starch and sucrose metabolism, cysteine and methionine metabolism, and cyanoamino acid metabolism. Furthermore, pentose and glucuronate interconversion, a pathway associated with cell wall metabolism, was significantly enriched during exposure to salt stress at 24 h and 48 h. Accordingly, this analysis suggests that the DEGs in the roots of *Aquilegia vulgaris* mainly respond to salt stress by participating in the aforementioned enriched pathway, and those involved in this pathway potentially regulate the salt tolerance of *Aquilegia vulgaris* roots.

### 2.7. Analysis of the Pentose and Glucuronate Interconversion Pathway

The KEGG enrichment analysis identified “pentose and glucuronate interconversion” as a significant enrichment pathway related to primary cell wall metabolism, so the combined GO and KEGG enrichment analysis focused on this pathway. In the salt stress treatment, this metabolic pathway consisted of 20, 12, and 11 DEGs at 12, 24, and 48 h, respectively. It mainly included Pectinmethylesterase (PME), Pectate Lyase Like (PLL), Polygalacturonas (PG), UDP-glucose pyrophosphorylase (UGP), and UDP-glucose dehydrogenase (UGD). The expression pattern analysis indicated (Figure 6) an up-expression trend throughout all stages of salt stress for *PME*: *gene-AQUCO_00700353v1*, *gene-AQUCO_00100227v1*, *gene-AQUCO_01300689v1*, and *gene-AQUCO_01300688v1*, while the remaining genes were downregulated. For *PLL*, *gene-AQUCO_03000036v1* and *gene-AQUCO_02700219v1* were upregulated at 12 h and 48 h, respectively, whereas the remaining genes were downregulated at all stages. For *PG*, *gene-AQUCO_01400346v1* was downregulated at 12 h, and *gene-AQUCO_03700071v1* was upregulated at all stages. All *UGP* were downregulated at 12 h and 24 h, but upregulated at 48 h. Additionally, *UGD* was consistently downregulated, while most genes in the pathway showed a trend of downregulated expression.

### 2.8. Network Analysis of Correlations between TFs and Functional Genes Related to the Cell Wall

The study found that the AP2 family had the most significant number of TFs, followed by MYB and bHLH. Relevant studies have proven that AP2, MYB, and bHLH could regulate downstream genes or create complexes to be involved in synthesising and modifying a plant cell wall [17]. The genes that were enriched in pentose and glucuronide interconversion pathways are primarily associated with the modification and synthesis of plant pectin in plant cell walls. Thus, we have constructed a possible regulatory network centred on functional genes that are associated with TFs and the cell wall. The results indicate (Figure 7) that there were 86 TF genes that had regulatory relationships with 21 functional genes. The genes *PG*, *PLL*, *PME*, *UGP*, and *UGD* might be directly regulated by AP2s, MYBs, and bHLHs. Furthermore, genes that are predominantly regulated by AP2s, MYBs, and bHLHs have been identified: *PLL*: *gene-AQUCO_01100377v1*, *gene-AQUCO_03400209v1*, *gene-AQUCO_10800001v1*, *gene-AQUCO_03800141v1*; *UGP*: *gene-AQUCO_02000159v1*; *PG*: *gene-AQUCO_03700071v1*; *PME: gene-AQUCO_00700353v1*, *gene-AQUCO_01300689v*, *gene-AQUCO_01000268v1*, and *gene-AQUCO_00100227v1*.

### 2.9. Validation of the RNA-Seq Data by qRT-PCR

To verify the accuracy of the sequencing results, this study selected nine genes for qRT-PCR analysis, which included seven genes from the pentose and glucuronide interconversion pathway, and two genes enriched in bHLH. The results (Figure 8) revealed that despite some differences in the up- or downregulation folds of expression between the transcriptome sequencing and qRT-PCR results, the gene expression patterns reflected by qRT-PCR were consistent with the transcriptome sequencing data. This further confirmed the high credibility and accuracy of the transcriptome sequencing data for *Aquilegia vulgaris* under salt stress and its ability to precisely reflect the response of the root of the plant to salt stress.

## 3. Discussion

Transcriptomics is uncovering the molecular regulatory mechanisms by which plants respond to salt stress through the transcription levels of plant mRNAs, which is critical for studying the molecular mechanisms of salt tolerance in plants [29]. In this study, 12 RNA-seq libraries were constructed for quality control analysis. The results indicate high-quality sequencing data that meet subsequent analysis standards. A larger number of Unigenes and more informative data, compared with previous studies on the transcriptome of ornamental plants [30], were obtained. The statistical analysis of the DEGs showed that the number of DEGs was highest at 12 h of salt treatment, and there was a gradual decrease in their number with the increase in stress time. Osmotic stress caused by salt stress is generally rapid but short-lived in plants, and this could explain the higher abundance of DEGs at 12 h of stress treatment [31]. Related studies have indicated that AS events are a key regulatory mechanism for plants, and that a number of proteins produced by specific AS transcripts are associated with salt stress responses in plants [32,33]. The study identified 307 DAS, and the emergence of DAS genes in *Aquilegia vulgaris* may be an energy-conserving strategy that is adopted to allow adaptation to adversity. Hence, DAS-related genes may have a critical role in adaptation to salt stress. In Zuo et al.’s [34] study on the transcriptome analysis of *Beta vulgaris* L. in response to alkaline stress, it was found that the main class of AS produced by DEGs was SE. The current study also found that the DAS types of all three treatment groups were dominated by SE, which is in agreement with Zuo et al.’s findings. This suggests that SE may play a crucial role in the acclimatisation of *Aquilegia vulgaris* to salt stress.

The GO enrichment analysis showed that cell wall in the CC classification and microtubule binding and cytoskeleton protein binding in the MF classification were significantly enriched in all treatment groups. Alterations in cell wall composition are a prominent aspect of the response to salt stress [35]. Salt stress affects the growth and development of plants by interfering with cell expansion and division, and the cell wall plays a key role in cell morphology and function. Salt stress results in water deficiency within plant cells, instigating alterations in cell expansion pressure [36]. The cell wall grants the mechanical resistance essential to withstand these cell expansion pressures [37,38]. The cytoskeleton’s structure is established through microtubules (MTs), microfilaments (MFs), and proteins that interact with both of them (MT/MF). The cytoskeleton of plants participates in several cellular processes, including extracellular and intracellular motility, cell division and growth, and membrane fixation. Due to the rapid polymerisation and depolymerisation of MT/MF, it is highly adaptable and restructured in plant cells to withstand abiotic stresses, including salt stress [13]. This analysis suggests that the structure and composition of the cell wall and the cytoskeleton may significantly affect the response of *Aquilegia vulgaris* to salt stress. There were 24 DEGs that were upregulated, while 23 downregulated DEGs were enriched in the cell wall. Additionally, 13 DEGs showed an upregulated pattern, while 46 DEGs showed a downregulated pattern enriched in microtubule binding. Dinneny [39] reported the downregulation of genes related to cell wall biosynthesis in epidermal and cortical cells, as well as the downregulation of microtubule-associated genes in cortical cells. Therefore, the downregulated DEGs enriched in the cell wall may play a role in cell wall biosynthesis. The DEGs enriched in microtubule binding were largely downregulated in expression, which is in agreement with Dinneny’s research.

The KEGG enrichment analysis revealed significant enrichment of the pentose and glucuronide interconversion pathway, related to cell wall metabolism, at 24 h and 48 h of salt treatment. The GO enrichment study suggests that cell wall dynamics may play a vital role in response to salt stress. Based on GO and KEGG analyses, attention was directed towards this pathway. In a study of transcriptional responses to different concentrations of nitrate in the root system of soybean (*Glycine max* L.) seedlings, Dai et al. [40] found that the pentose and glucuronide interconversion pathway was the most significantly enriched pathway in all three comparison groups. Similarly, Lei et al. [41] observed that the pathway is involved in the regulation of adventitious root formation in apple (*Malus x domestica* Borkh). Cao et al. [42] concluded that the pentose and glucuronide interconversion pathway is involved in the regulation of root epidermal differentiation and root hair tip growth in *Arabidopsis thaliana* under cadmium treatment. Additionally, Zhang et al. [43] showed that the pathway plays a role in maize root development. This pathway is anticipated to play a vital role in the adaptability of *Aquilegia vulgaris* root growth and development, as well as its resistance to abiotic stresses.

The key genes encompassed in this pathway are *PME*, *PG*, *PLL*, *UGP*, and *UGD*, all of which are linked to the synthesis and structure of pectin. In the current study, the three main pathways involved in the pectin synthesis pathway are the D-galacturonate pathway, the myo-inositol pathway, and the UDP-glucose pathway [44]. The pentose and glucuronide interconversion pathway primarily consists of the D-galacturonate pathway and the UDP-glucose pathway. The D-galacturonate pathway primarily comprises *PME*, *PLL*, and *PG*, whilst the UDP-glucose pathway comprises mainly *UGP* and *UGD*. Yan et al. [45] discovered that in *Arabidopsis thaliana*, the expression of *PME31* increased significantly under salt stress, indicating a positive regulatory role of *PME31* in salt stress tolerance. The study found that the *PMEs* in the D-galacturonate pathway, namely *gene-AQUCO_00700353v1*, *gene-AQUCO_00100227v1*, *gene-AQUCO_01300689v1*, and *gene-AQUCO_01300688v1*, demonstrated an upward trend in expression during all stages of salt stress. This increased expression may enhance the plant’s tolerance. Analysis of *Arabidopsis PLLs* by Renault et al. [46] demonstrated that *AtPLL26* and *AtPLL19* exhibited decreased expression upon salt stress treatment. Most of the *PLLs* in the D-galacturonate pathway were downregulated throughout all stages of expression, which may aid in plant resistance to salt stress. Yang et al. [47] found that two *PGs (PtPG58* and *PtPG67*) were specifically expressed in leaf abscission zones under salt stress in Populus, and two *PGs* were detected in the D-galacturonate pathway. They also found that *PG* may be involved in the response to salt stress. The salt regulation of *UGP* has been observed in several species or varieties, yielding varying results. For example, salinity has been shown to reduce the expression levels of *UGP* in the root of rice (*Oryza sativa* L. cv. Nipponbare) [48]. In comparison, in barley (*Hordeum vulgare*), tolerant *UGP* expression levels were upregulated in the root when compared to the sensitive strain [49]. Additionally, the expression of *UGD* was found to be downregulated in tolerant barley [50]. Both *UGPs* in the UDP-glucose pathway were downregulated at 12 and 24 h but upregulated at 48 h, and *UGD* genes were downregulated at all stages. This suggests that the reduction in the expression of *UGP* and *UGD* in the root of *Aquilegia vulgaris* could contribute to enhancing its salt tolerance.

Transcriptional regulation is a vast cell wall regulatory mechanism that simultaneously influences multiple metabolic pathways and orchestrates the transcription of numerous genes related to the cell wall. Cell wall regulation occurs through a hierarchical transcriptional network of several TFs [51]. Meanwhile, TFs are deemed crucial controllers of target gene expression in plants, influencing salt tolerance levels [52]. Multiple studies have shown that TFs, including AP2, MYB, bHLH, NAC, WRKY, and C2H2, can respond to salt stress by regulating downstream genes. They are essential for plant defence against salt stress [53,54,55,56]. The family with the highest number of TFs in this study was AP2, followed by MYB and bHLH. These may be the primary regulators of the *Aquilegia vulgaris* response to salt stress. Ding et al. [17] examined the link between pectin methyl esterification modifications and TFs and noted that *ERF4* primarily regulated *PMEI13* and *PMEI15* via transcriptional repression, whilst *MYB52* indirectly upregulated the expression of *PMEI13* and *PMEI15* by counteracting the DNA-binding ability of *ERF4*. According to Sun et al. [57], bHLH activates *GhPLL76* expression by binding to the G-box located within the promoter region of *GhPLL76*. A potential regulatory network was constructed in this study, centred on functional genes related to TFs and the cell wall. The results indicate that AP2, MYB, and bHLH may directly regulate *PG*, *PLL*, *PME*, *UGP*, and *UGD*. Accordingly, it is predicted that AP2, MYB, and bHLH may regulate the plant cell wall through the regulation of the downstream genes *PG*, *PLL*, *PME*, *UGP*, and *UGD*, which in turn affect the salt tolerance of the plant. Further investigation of the specific modes of regulation is required. In this study, important salt-stress-related genes and metabolic pathways were tentatively predicted in *Aquilegia vulgaris*. Potential regulatory relationships between functional genes and TFs in significant pathways were established. This study’s findings provide valuable genetic resources for enhancing salt tolerance in *Aquilegia vulgaris* and selecting and breeding novel salt-tolerant varieties using genetic engineering technology [8].

## 4. Materials and Methods

### 4.1. Treatment of Plant Material with Salt

Test materials comprised *Aquilegia vulgaris* seedlings. Seeds of *Aquilegia vulgaris* were acquired from the Ornamental Plant Resources Research and Germplasm Resources Innovation Team at the College of Horticulture, Jilin Agricultural University (125°43′ E, 43°82′ N), Jilin, China. The seeds were planted in 5.8 × 5.8 × 11 cm^3^ pots filled with a mixture of charcoal (Wangda charcoal soil sales base, Changchun, China), garden soil (Wangda charcoal soil sales base, Changchun, China), and perlite (Senxin Perlite Factory, Changchun, China) in a 3:1:1 ratio and then placed in the greenhouse at Jilin Agricultural University for germination and growth. The incubation conditions comprised a temperature of 28 ± 3 °C and humidity of 40% during the day, and a temperature of 23 ± 2 °C and humidity of 50% during the night, with natural lighting conditions. When the annual seedlings reached six true leaves, those demonstrating uniform growth were chosen as the test materials for salt stress treatment. Based on the preliminary experiment, it was established that each container was filled with 60 mL of a 200 mM sodium chloride solution [24]. The whole roots were collected at 12, 24, and 48 h after treatment, using 0 h as the control. The collected samples were rapidly frozen in liquid nitrogen and preserved at −80 °C in a refrigerator. Each sample was replicated three times, with three plants in each replication.

### 4.2. Total RNA Extraction and cDNA Library Construction from the Roots of Aquilegia Vulgaris

We extracted total RNA from the roots using the RNApure Plant Kit (Kangwei Century, Beijing, China). Then, we conducted rigorous quality control of the RNA samples by testing their integrity with an Agilent 2100 bioanalyser (Agilent, Santa Clara, CA, USA). Following sample qualification, library construction took place, whereby mRNA with polyA tails was enriched by Oligo (dT) magnetic beads. Subsequently, cDNA was synthesised using fragmented mRNA as a template. Libraries were obtained after undergoing end repair, the addition of A tails, the ligation of sequencing junctions, and PCR amplification using specific primers. Once the libraries passed the inspection, they were combined based on their effective concentration and desired downstream data volume. The combined libraries were then subjected to Illumina (LC Sciences, San Diego, CA, USA) sequencing using *Aquilegia coerulea* as a reference.

### 4.3. Analysis of Transcriptome Data

The genome sequence and genome annotation files for *Aquilegia coerulea* were sourced from the online website NCBI (https://ftp.ncbi.nlm.nih.gov/genomes/all/GCA/002/738/505/GCA_002738505.1_Aquilegia_coerulea_v1/ (accessed on 1 December 2021). HISAT2 2.2.1 (Johns Hopkins University, Baltimore, MD, USA) software was used to align the clean reads with the reference genome in order to gather information on their position [58]. Quantitative gene expression levels for each sample were analysed separately, and then combined to obtain the expression matrix for all samples. FPKM values, calculated using StringTie 1.3.4 (Johns Hopkins University, Baltimore, MD, USA) software, were used to quantify expression abundance and changes [59,60]. DESeq2 4.1.1 (Harvard T.H. Chan School of Public Health, Cambridge, MA, USA) software was used for the differential expression analysis, with |log2(FoldChange)| > 0 and padj ≤ 0.05 as the screening criteria [61,62]. The DEGs were compared with the PlantTFDB database in order to obtain the TF classification and the number of DEGs [31]. The genes were analysed using clusterProfile 4.2.1 (Jinan University, Guangdong, China) software for GO and KEGG enrichment. A *p*-value ≤ 0.05 was selected as the threshold to indicate significant enrichment for both GO and KEGG [63]. Quantitative and differential analyses of AS events were carried out using the rMATS 4.0.1 (University of California, Los Angeles, CA, USA) software. The rMATS 4.0.1 (University of California, Los Angeles, CA, USA) software was used to categorise AS events into five categories, and differential AS was analysed in samples with biological replicates [33]. The correlation among the samples was determined using the Pearson correlation (R^2^) coefficient [64]. The correlation network between groups was mapped using the Metware online tool available at https://cloud.metware.cn/#/home (accessed on 10 March 2023) [65]. The results were then visualised with the help of Cytoscape 3.9.0 (National Institute of General Medical Sciences, Bethesda, MD, USA).

### 4.4. Validation of Gene qRT-PCR

RNA extraction from the root of *Aquilegia vulgaris* was carried out using the RNApure Plant Kit (Kangwei Century, Beijing, China). cDNA synthesis was performed using the reverse transcription kit (Novoprotein, Shanghai, China). Specific primers were designed using Primer 5.0 (Primier Biosoft International, Palo Alto, CA, USA) software. The internal reference gene, *IPP2* [66], was chosen, and the primer sequences are listed in Appendix A. The qRT-PCR reaction was conducted with the NovoScript^®^ SYBR qPCR SuperMix Plus kit (Novoprotein, Shanghai, China). The reaction procedure and reaction system were interpreted according to the kit instructions. To ensure the reliability of the measurements, we performed three biological replicates and two technical replicates for each sample. The relative gene expression was calculated using 2^−ΔΔCT^ [67].

## 5. Conclusions

This study conducted transcriptome sequencing on the root of *Aquilegia vulgaris* seedlings subjected to NaCl for 0, 12, 24, and 48 h to simulate salt stress. According to GO enrichment results, the composition and structure of the cell wall and cytoskeleton may significantly influence the response to salt stress. KEGG annotation indicated that the pentose and glucuronate interconversion pathway associated with cell wall metabolism was significantly enriched at 24 h and 48 h of salt treatment. The outcomes of GO and KEGG enrichment indicate that the cell wall likely plays a crucial role in the reaction of *Aquilegia vulgaris* roots to salt stress. The large number of DEGs that are enriched in the TF families of AP2, MYB, and bHLH indicate that they could potentially serve as crucial regulators in the response of *Aquilegia vulgaris* to salt stress. The correlation network analysis predicted the regulatory network of TFs and functional genes related to the cell wall. The results indicate that AP2, MYB, and bHLH may regulate *PG*, *PLL*, *PME*, *UGP*, and *UGD*. In this study, a time-course transcriptomic analysis of *Aquilegia vulgaris* roots was executed to investigate the response to salt stress. The response network of *Aquilegia vulgaris* roots to salt stress was preliminarily explored, highlighting an important resource for mining salt tolerance genes. Furthermore, this research lays a significant foundation for the further investigation of gene interactions and the systematic study of the salt tolerance mechanism in *Aquilegia vulgaris*.

## Figures and Tables

**Figure 1 ijms-24-16450-f001:**
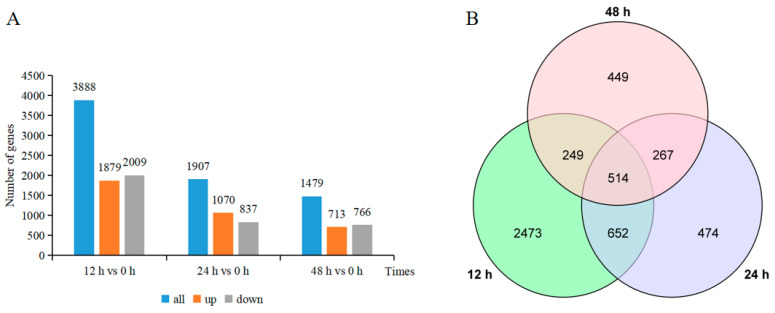
Number and distribution of DEGs in *Aquilegia vulgaris* roots. (**A**) The number of DEGs that were upregulated or downregulated at the three time points of NaCl stress. (**B**) Venn diagram showing DEGs overlapping at the three time points of NaCl stress or DEGs unique to each NaCl stress time points. The overlapping part is the number of co-induced genes.

**Figure 2 ijms-24-16450-f002:**
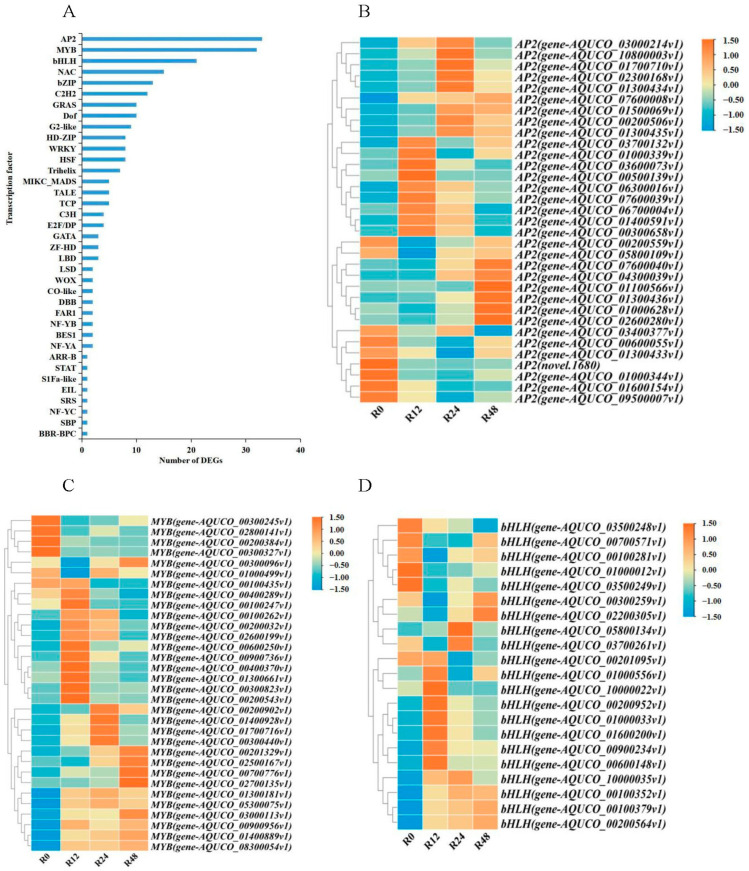
Enrichment and expression pattern analysis of TF families to which DEGs belong. (**A**) Classification of TF families for DEGs. (**B**) Heat map of *AP2* expression. (**C**) Heat map of *MYB* expression. (**D**) Heat map of *bHLH* expression. The abscissa displays the processing times of the various samples, while the ordinate represents information on the genes within each TF family. The diagram illustrates the log 2 fold change values of every gene in the treated and untreated groups, with a colour gradient ranging from orange to blue indicating changes in DEG expression levels.

**Figure 3 ijms-24-16450-f003:**
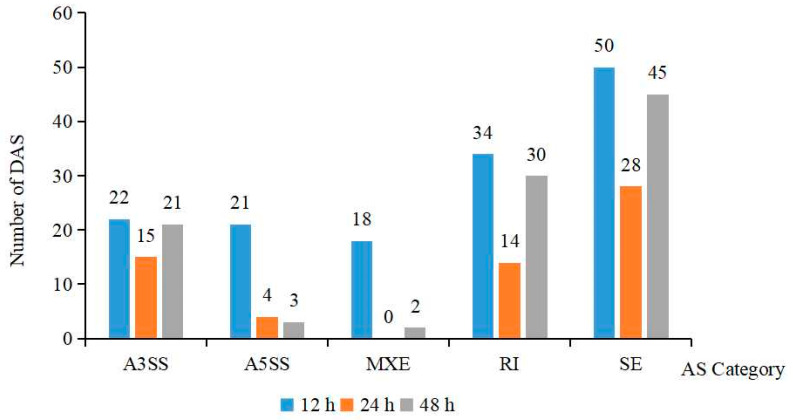
DAS analysis of DEGs in the root of *Aquilegia vulgaris* under NaCl stress.

**Figure 4 ijms-24-16450-f004:**
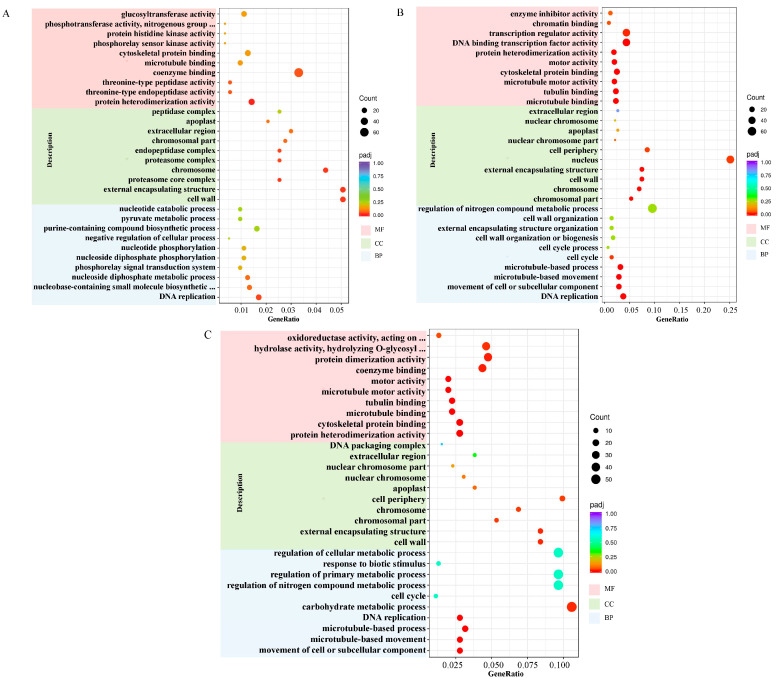
GO enrichment analysis of DEGs. (**A**) Analysis of GO enrichment in NaCl stress for 12 h. (**B**) Analysis of GO enrichment in NaCl stress for 24 h. (**C**) Analysis of GO enrichment in NaCl stress for 48 h. The dots’ size and colour signify gene quantity and the *p*-value, respectively (with red to purple indicating small to large). Different colours indicate various functional classifications.

**Figure 5 ijms-24-16450-f005:**
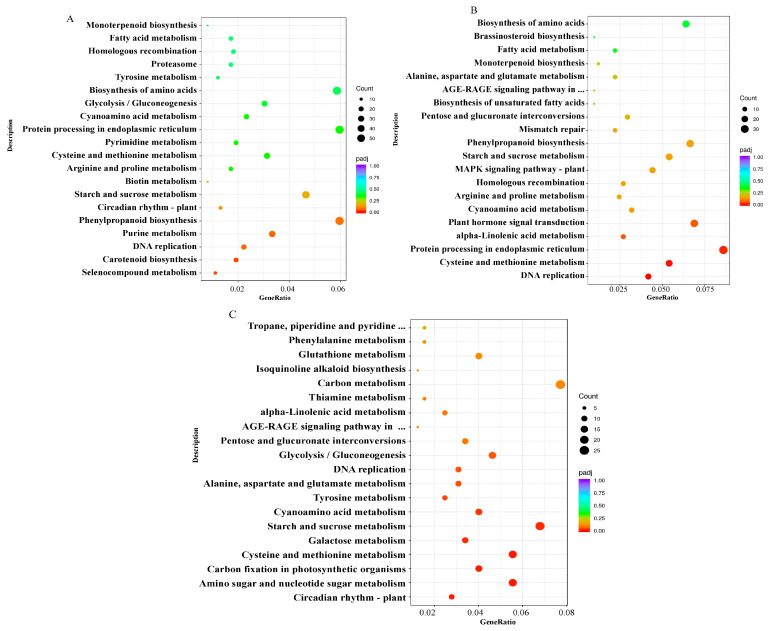
KEGG enrichment analysis of DEGs. (**A**) Analysis of KEGG enrichment in NaCl stress for 12 h. (**B**) Analysis of KEGG enrichment in NaCl stress for 24 h. (**C**) Analysis of KEGG enrichment in NaCl stress for 48 h. The dots’ size and colour signify gene quantity and the *p*-value, respectively (with red to purple indicating small to large).

**Figure 6 ijms-24-16450-f006:**
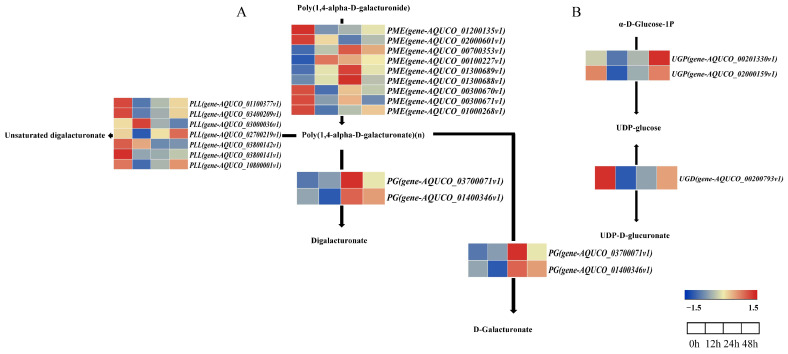
Analysis of gene expression involved in the pentose and glucuronate interconversion pathway. (**A**) The D-galacturonate pathway. (**B**) The UDP-glucose pathway. The colour gradient ranging from red to blue reflects alterations in the expression levels of DEGs.

**Figure 7 ijms-24-16450-f007:**
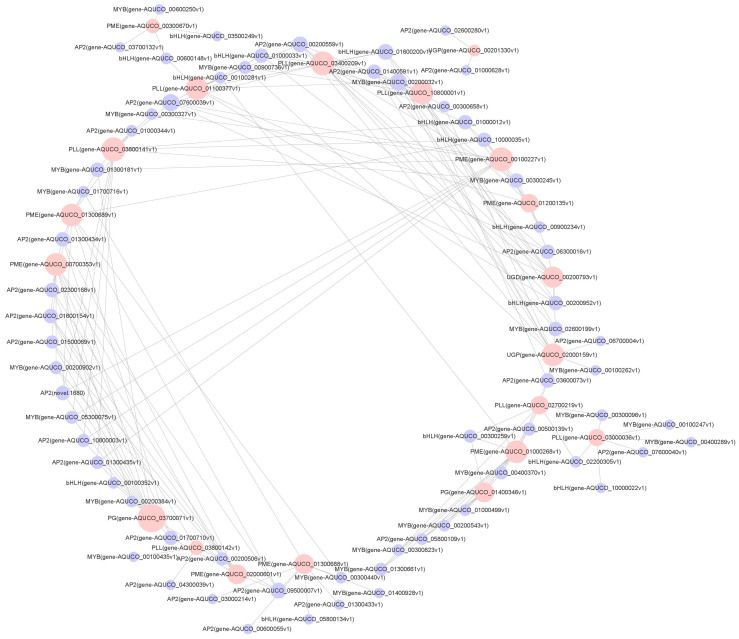
Network analysis of functional genes in the pentose and glucuronate interconversion pathway in correlation with AP2, MYB, and bHLH TFs. Pink circles represent functional genes, purple circles represent TF families, and the connecting lines indicate potential co-expression relationships between TFs and functional genes. The diameters of the nodes represent the interaction frequency.

**Figure 8 ijms-24-16450-f008:**
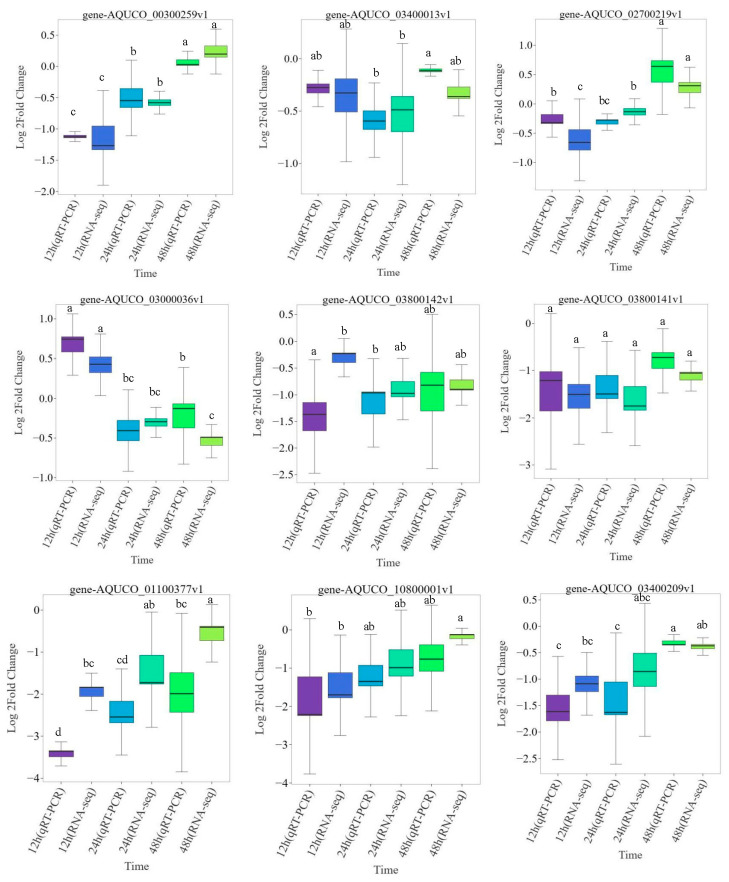
The expression patterns of the nine selected genes in roots were verified by qRT-PCR. The time and data type are plotted on the abscissa, and the log 2 fold change value is shown on the ordinate. Groups without the same letter exhibited statistically significant differences (*p* < 0.05).

## Data Availability

The data presented in this study are openly available in the Sequence Read Archive (SRA) database in NCBI under BioProject numbers PRJNA1035280.

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
