# Peer review of "Time-Course Transcriptome Analysis of Aquilegia vulgaris Root Reveals the Cell Wall’s Roles in Salinity Tolerance"

_ijms, 2023, doi:10.3390/ijms242216450_

Round 1

Reviewer 1 Report

Comments and Suggestions for Authors

The authors investigate salt stress in Aquilegia vulgaris, a salt-tolerant plant. Researchers analyzed gene expression in its roots under salt treatment and identified differentially expressed genes. They found that transcription factors like AP2, MYB, and bHLH may be linked to salt tolerance. The cell wall and the pentose and glucuronate interconversion pathway play essential roles in the plant's response to salt stress. This research provides insights into salt tolerance mechanisms and potential applications in developing salt-resistant Aquilegia vulgaris cultivars.

Overall, the study's objectives are suitable for publication. The experiments were well-executed. However, the data analyses related to the transcriptome were not described in detail. Tables and figures were inadequately prepared, and detailed legends for them are necessary. Raw data were not deposited in a public database, such as the SRA database in NCBI. The manuscript contains numerous errors, and it appears rushed without careful revisions. Therefore, I recommend rejecting the manuscript. Nevertheless, I encourage the authors to resubmit it after making rigorous revisions.

Several issues need attention: There are no line numbers, making it difficult for reviewers to provide comments. Spell out all abbreviations upon their first use, e.g., TFs, DEGs, GO, KEGG, etc. Insert a space between a sentence and a reference, e.g., "encounter [1-2]." Table 1 could be provided as a supplementary table, and it's recommended to include a table with detailed sample information, including tissue, time point, replicate number, and accession numbers. Provide complete details for each table and figure in their legends. Figure 4 is not legible; please increase the font size and revise the figure legends. The paragraph related to transcription factor genes should be rewritten, as should the one about alternative splicing. Figure 5 lacks detailed legends, and the font sizes in figures 6, 7, 8, 9, and 10 are too small, necessitating improvements. Specify the values used for DESeq2 to identify DEGs, and review the screening criteria for DEGs (|log2(FoldChange)| > 0 and padj ≤ 0.05) as they appear unusual. Finally, add references for the software used in data analysis and deposit raw data in a public database, including accession numbers in the manuscript.

Comments on the Quality of English Language

 Extensive editing of English language required.

Author Response

Dear reviewer,

We are really appreciate for your excellent and professional revision of this manuscript. We have checked the manuscript according to the comments. After carefully studying, we have made corresponding changes on the manuscript and uploaded to the attachment. Hope these will make it more acceptable for publication.

If any other information or modification are needed, please let me know, thank you so much.

Yours sincerely,

Yun Bai

Reviewer 2 Report

Comments and Suggestions for Authors

In my opinion, the root transcriptome study of Aquilegia vulgaris in response to salt stress is interesting. In general, the study seems to be well done. However, I think that the material and methods section could be more detailed, also at the level of experimental design. I think that more background examples of results from other halophytes could be included and the possible applications of this study in the induction of salt resistance in cultivated plants could be discussed. The legends of the tables and figures should be improved by including all the details that make them understandable, as the handwriting of some figures is illegible. The format and the English should be reviewed. Further comments can be found in the attached document.

Comments on the Quality of English Language

In my opinion, the manuscript should be reviewed by a native English speaker.

Author Response

(The authors gave the same response as above.)

Round 2

Reviewer 1 Report

Comments and Suggestions for Authors

The authors have diligently incorporated revisions based on the reviewer's comments. I wholeheartedly recommend this manuscript for publication in its present state. Congratulations on the outstanding work.